# RTLSeek: Boosting the LLM-Based RTL Generation with Multi-Stage Diversity-Oriented Reinforcement Learning

## Abstract

Register Transfer Level (RTL) design translates high-level specifications into hardware using Hardware Description Language (HDL) like Verilog. While LLM-based RTL generation holds promise, limited functionally verifiable high-quality data constrains its accuracy and diversity. Current post-training generates one-to-one HDL code from natural language, lacking deep understanding of RTL variations for different goals. This paper proposes RTLSeek, a novel post-training paradigm that employs rule-based Diversity-Oriented Reinforcement Learning to improve RTL accuracy and diversity. We introduce a Diversity-Centric Multi-Objective Reward Scheduling that integrates expert knowledge and EDA feedback, along with a three-stage training framework to better utilize scarce data. Experiments show RTLSeek outperforms other methods on RTLLM benchmark, with ablation studies validating that our approach enhances the final quality of generated RTL by encouraging a wider exploration of the design space and achieving "the more generated, the better results". We release RTLSeek's complete implementation framework, including the dataset, source code, and model weights, at `https://anonymous.4open.science/r/ICLR24081-9C28/`.

## 1 Introduction

Register Transfer Level (RTL) design is the process of converting high-level functionality descriptions into the topology of circuit functional blocks via Hardware Description Language (HDL) like Verilog. It is the most complex and wide-ranging stage of the digital circuit customization flow. RTL coding is not well automated, and its quality still relies heavily on the experience of engineers. Recently, the development of Large Language Models (LLMs) (Brown et al., 2020; Ouyang et al., 2022; Achiam et al., 2023) has advanced considerably, enabling automatic code generation for various programming languages. The potential of using LLMs to automatically generate RTL code is an exciting area of research, with pioneering studies suggesting the feasibility of LLMs to align human language with RTL design languages and their functional specifications (Chang et al., 2024; Chen et al., 2024; Liu et al., 2023a; Thakur et al., 2023; Liu et al., 2023b; Lu et al., 2023). Despite these gains, current research on RTL generation remains inadequate. Even on relatively simple test suites, the accuracy of generated code remains low (Liu et al., 2025). One of the main bottlenecks is the scarcity of high-quality training data. Specifically, only around a thousand of samples with testbenches, which are essential for verifying the correctness of model-generated codes, are available by now. From an academic perspective, it seems unlikely that the open-sourcing of high-quality chip design data will be resolved in the near future. In light of these constraints, one of the key questions for the LLM RTL design community is **how can we harness circuit design experience to improve training, without simply waiting for more open-source, functionally verifiable high-quality datasets?**

Recalling how human learners tackle complex tasks such as RTL design, they often employ an approach akin to "**learning by seeking**": for a single problem, some advanced learners will try some different solutions, validating their correctness along the way. This process allows them to extract more information from a limited training set and gain a deeper understanding of the problem-solving methodology. The observation from Fig. 1a demonstrates that Test Time Scale (TTS)-based prompting for diversity could help in successful RTL generation. This finding directly motivates

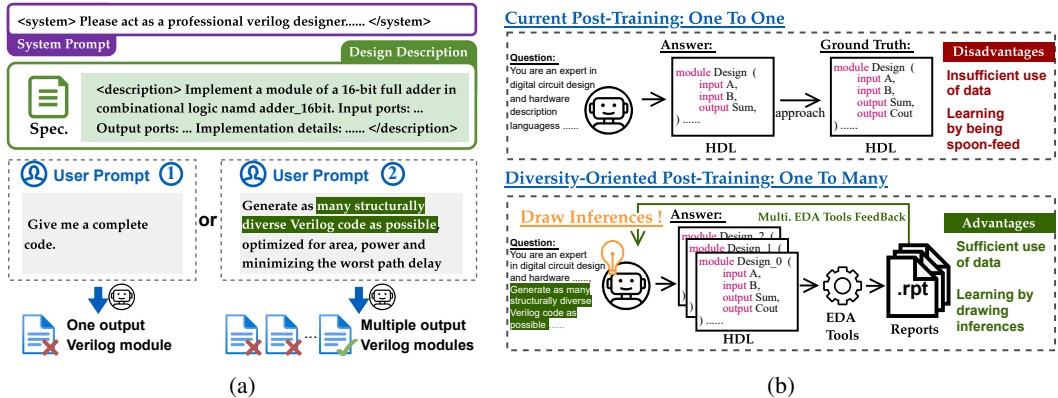

Figure 1: (a) Observation from GPT-4's RTL design: achieving Test Time Scale (TTS) (Zhang et al., 2025) by designing a multi-sample user prompt enhances the correctness of generated RTL syntax and functionality. (b) Motivation of RTLSeek: the diversity-oriented post-training could well overcome the limitations of insufficient training data.

us to explore a more fundamental solution: enhancing the model's intrinsic capability during post-training to reason about and generate diversity, achieving **"the more generated, the better results"**. However, as shown in Fig. 1b, for LLM training, the current widely adopted one-to-one post-training paradigm (Zhu et al., 2025) more closely resembles rote memorization, requiring the LLM to merely memorize the answer rather than actively reason about the principles underlying RTL design. Drawing inspiration from human study methods, if we encourage the LLM, during the RTL coding training process, to freely explore with diversity-oriented for the same design description by generating code with different structures and then evaluating them, we can make more effective use of high-quality testbench-equipped datasets and foster a deeper understanding of RTL development. However, implementing this diversity-oriented approach is not trivial. Two key challenges arise: (1) Designing an efficient training framework: How to effectively explore diverse RTL solutions when only a limited dataset is available. (2) Balancing diversity and correctness: how to ensure the model maintains rigorous correctness while seeking a broad range of designs.

In this paper, we introduce **RTLSeek**, a Diversity-Oriented Reinforcement Learning (RL) post-training framework tailored to LLM-based RTL code generation. Our contributions are summarized as follows: (1) By integrating Group Relative Policy Optimization (GRPO) into the training framework, RTLSeek systematically explores RTL code diversity during post-training, resulting in significant improvements in RTL code generation capabilities. (2) To balance diversity and correctness, RTLSeek features a Multi-Objective Reward Scheduling, incorporating expert IC design constraints, EDA tool feedback, and dynamic evaluation of the quality of the current generated result in intermediate step during training. (3) To overcome the challenge of insufficient verifiable datasets, RTLSeek adopts a three-stage training framework: ① warm-up using Supervised Fine-Tuning, ② diversity-oriented exploration, and ③ multi-objective-oriented exploration, striking an optimal balance between exploration and correctness. (4) Experimental results show that our post-training paradigm improves Qwen 2.5's RTL generation performance by over 40%, surpassing other methods. Ablation studies confirm that removing diversity rewards or any training stage degrades performance, highlighting the effectiveness of our diversity-oriented RL approach in data-scarce RTL tasks.

## 2 DESIGN OF RTLSEEK

In this section, we present the whole design framework of RTLSeek as shown in Figure 2. First, we formalize the problem formulation for LLM-based RTL generation and outline the overall design of RTLSeek. Next, we detail our hybrid training paradigm that integrates SFT with GRPO-based reinforcement learning, followed by the multi-objective reward mechanism that dynamically balances correctness and diversity. Our diversity quantification is based on Abstract Syntax Tree (AST)-based structural analysis approach which is shown in Appendix A.1.

Figure 2: The three-stage hybrid training paradigm in RTLSeek.

## 2.1 PROBLEM DEFINITION

Our problem can be defined as follows. Given a natural language specification $S$ describing desired hardware behavior (see *user prompt* in Figure 3), the RTL generation task aims to produce a set of design implementations $\{D_1, D_2, ..., D_n\}$ satisfying:

1. **Functional Equivalence**: $\forall D_i, D_j \in \{D\} : D_i \equiv D_j$ where $\equiv$ denotes behavioral equivalence verified through testbench.

2. **Structural Diversity**: $\text{Struct}(D_i) \neq \text{Struct}(D_j)$ for $i \neq j$, where $\text{Struct}(\cdot)$ represents the Abstract Syntax Tree (AST) representation.

3. **Non-trivial Variation**: $\text{Diversity}(\text{Struct}(D_i), \text{Struct}(D_j)) > \delta$, where $\delta$ is the minimum variation threshold excluding superficial changes (e.g., variable renaming).

## 2.2 OVERALL DESIGN

This paper addresses a critical problem in RTL generation using LLMs: how to effectively leverage existing circuit design expertise to enhance model training, given the severe scarcity of high-quality, verifiable training data. The current landscape presents a fundamental limitation that only several thousand samples with accompanying testbenches (essential for validating model-generated RTL code) are publicly available. This data scarcity creates a pressing need for alternative training paradigms that can compensate for the lack of large-scale, high-quality datasets.

Existing solutions fall short in two key aspects. First, verification-based RTL generation methods (Sami et al., 2024; Huang et al., 2024) achieve only superficial diversity improvements (e.g., simple variable renaming) at high computational cost. Second, while SFT ensures basic correctness, it fails to produce sufficiently diverse designs; conversely, pure diversity-driven approaches often generate non-functional code. This reveals a fundamental tension in RTL generation: how to simultaneously ensure functional correctness while exploring meaningful design variations under tight data constraints.

To address these challenges, we propose RTLSeek, a GPRO-based training paradigm that combines three key innovations: (1) A hybrid training paradigm integrating SFT with two-stage GRPO optimization (as shown in Figure 2); (2) An automated reward system using circuit analysis tools to balance correctness and diversity (as shown in Figure 3); (3) AST-based structural equivalence analysis to quantify and encourage design diversity (as shown in Figure 5).

Unlike previous methods that treat correctness and diversity as competing objectives, RTLSeek's integrated approach enables efficient exploration of the design space while maintaining functional validity, even with limited training data.

## 2.3 THE HYBRID TRAINING PARADIGM

In this section, we introduce the hybrid training paradigm integrating the SFT and the two-stage GRPO-based RL. We first present the preliminary of the GRPO-based RL. Then we give the details of the training paradigm.

**GPRO-based Reinforcement Learning**.

GRPO extends policy gradient methods by introducing group-wise relative comparisons, addressing critical limitations in conventional RL approaches for generative tasks. Originally demonstrated in

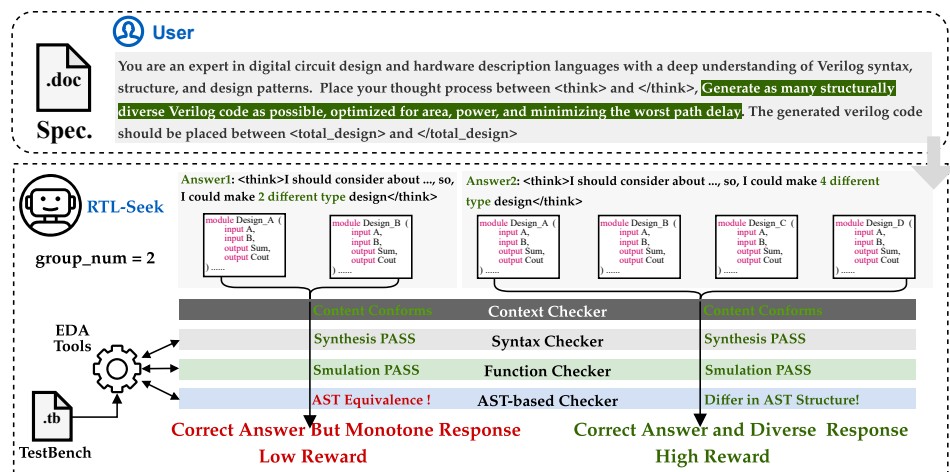

Figure 3: Diversity-Centric Multi-Objective Reward reinforcement learning in RTLSeek.

DeepSeek R1(Guo et al., 2025), GRPO's unique characteristics make it particularly suitable for RTL generation: (1) **Stabilized Group Updates**: By constraining policy updates within output groups through KL-divergence regularization, GRPO mitigates the mode collapse problem prevalent in PPO while maintaining exploration efficiency. The group-wise mechanism provides more reliable gradient estimates compared to single-sample methods. (2) **Relative Quality Evaluation**: GRPO's novel optimization surface considers relative rankings within output groups, contrasting with PPO's absolute advantage estimation and DPO's pairwise comparisons. This enables joint evaluation of functionally equivalent but structurally diverse RTL designs.

Formally, given a question $q$, GRPO samples a group of outputs $\{o_1, o_2, ..., o_G\}$ from the old policy $\pi_{\theta_{\text{old}}}$ and optimizes the new policy $\pi_\theta$ by maximizing the objective function:

$$
\mathcal{J}_{\text{GRPO}}(\theta) = \mathbb{E}\left[q \sim P(Q), \{o_i\}_{i=1}^G \sim \pi_{\theta_{\text{old}}}(O[q])\right] \frac{1}{G} \sum_{i=1}^{G} \left( \min\left( \frac{\pi_\theta(o_i|q)}{\pi_{\theta_{\text{old}}}(o_i|q)} A_i, \mathcal{C} \right) - \beta \mathbb{D}_{KL}(\pi_\theta \| \pi_{ref}) \right),
$$

$$
\mathcal{C} = \text{clip}\left( \frac{\pi_\theta(o_i|q)}{\pi_{\theta_{\text{old}}}(o_i|q)}, 1 - \varepsilon, 1 + \varepsilon \right) A_i,
$$

(1)

where $\pi_\theta$ is the current policy, $\pi_{\theta_{\text{old}}}$ is the old policy, $A_i$ is the advantage of the i-th output, and $\varepsilon$ is the hyperparameter used to control the update amplitude of the policy. $\beta$ is KL divergence penalty term coefficient, $\mathbb{D}_{KL}(\pi_\theta \| \pi_{ref})$ is KL divergence between the strategy $\pi_\theta$ and the reference strategies $\pi_{ref}$. Figure 3 illustrates the group comparison mechanism with $G = 2$, showing how relative evaluations enable diverse solution generation.

**Training Paradigm**.

As shown in Figure 2, to maximize the utility of limited training data, we develop a hybrid training paradigm combining SFT with a two-stage GRPO optimization process. The complete training scheme enables simultaneous optimization for both correctness and diversity, overcoming the limitations of conventional approaches that typically prioritize one aspect at the expense of the other. The pipelines of the three stages are detailed as follows (Shown in Figure 2).

Stage 1. SFT Warm-up: Learn what is HDL. We first train on curated RTL datasets containing both verified and unverified examples from reputable open-source projects. These natural language-to-Verilog pairs establish fundamental syntax and functional understanding, stabilizing later RL training. The SFT phase only requires a single Verilog module output for a query and focuses on learning basic code patterns and common design constructs, while filtering out low-quality examples through careful dataset curation. This initialization is crucial as it provides a strong starting point for subsequent reinforcement learning, preventing the model from exploring invalid design spaces during RL training.

Stage 2. GRPO: Learn to generate diverse HDLs. Using large-scale unverified datasets, this phase employs diversity rewards to: (1) maximize data utility and generalization, while (2) developing varied yet syntactically valid RTL generation capability. We specifically design the diversity reward to encourage structural variations in control logic, datapath organization, and module hierarchy. The group-based sampling in GRPO allows the model to compare multiple design alternatives simultaneously, learning to generate different implementations for the same specification. This phase significantly expands the model's design repertoire beyond what could be learned through SFT alone.

Stage 3. GRPO: Learn to generate right and diverse HDLs. The final stage applies combined diversity and functional rewards on verified datasets. This refines the model's functional-code correspondence understanding, requiring careful parameter calibration from previous phases. The testbench verification provides precise feedback for functional correctness, while the diversity component maintains the variation learned in previous stages. This two-tiered reward structure ensures the model generates both correct and innovative designs.

## 2.4 MULTI-OBJECTIVE REWARD SCHEDULING

The key challenge in improving the exploration quality of LLMs in RL lies in designing an appropriate reward mechanism that both refines decisions and provides systematic guidance throughout the learning process.

Traditional post-training paradigms based on SFT typically utilize syntax and functional correctness feedback from EDA tools (Chang et al., 2024) to refine reference responses. However, these approaches often underutilize scarce training datasets and currently lack effective verification methods to ensure functional correctness for most testbench-free datasets.

As illustrated in Fig 3, we propose Multi-Objective Reward Scheduling. This approach not only incorporates syntax and functional correctness feedback but also integrates a diversity reward obtained through Abstract Syntax Tree (AST) analysis to guide LLMs in generating structurally diverse RTLs. Additionally, we introduce a context reward to regulate reasoning quality, thereby achieving an effective exploration-exploitation balance.

The whole reward function $R_{total}$ is as follows:

$$R_{total} = R_{syn} + R_{func} + R_{div} + R_{cont}, \tag{2}$$

Here, $R$ epresents the reward functions we have designed separately. By decomposing and combining different rewards for various RL training stages, we are able to maximize the exploration capabilities of the LLM within the constraints of limited training data. These components are explained in detail below.

**Syntax Correctness Reward**.

$R_{syn}$ is obtained by verifying whether the generated Verilog code adheres to the syntactic rules of the Verilog language, using the syntax analysis tool, Pyverilog (Takamaeda-Yamazaki, 2015a): if any RTL in the generated set passes the syntax check, then $R_{syn} = 1$; otherwise, $R_{syn} = 0$.

**Function Correctness Reward**.

$R_{func}$ aims to guide the LLM in generating functionally correct RTL code. Only a small subset of datasets have a testbench verification set, which is used with the commercial simulation tool VCS to simulate the generated RTL. If any RTL in the generated set passes the testbench simulation, $R_{func} = 1$; otherwise, $R_{func} = 0$.

**Diversity Reward**.

Since different RTL implementations can exist for the same functional specification under varying design objectives in IC design, we expect the LLM to explore as many diverse design solutions as possible, rather than repeatedly generating the same design. To achieve this goal, we propose $R_{div}$ as follows:

$$R_{div} = N_c + N_s, \tag{3}$$

where $N_c$ and $N_S$ represent the number of generated RTL codes with structural diversity that pass syntax check and functional verification, respectively.

**Context Reward**.

$R_{cont}$ consists of two components: format correctness and reasoning length reward. The format correctness requires the model to enclose the reasoning process of the given problem within the $< think >< /think >$ tags, and to place the final multiple RTL solutions within the $< total\_design >< /total\_design >$ tags. If the output match the format, a reward of 0.5 will be given; otherwise, a penalty of -0.5 will be given. The reasoning length reward is obtained from the significant correlation between the length of the reasoning response and the quality of the generated output (Yeo et al., 2025). Specifically, if the reasoning response is too short, the model may fail to fully explain the reasoning steps, resulting in reduced accuracy and incompleteness in the generated response. Conversely, when the reasoning response is too long, the model may produce redundant or irrelevant content or become entangled in unnecessary, lengthy reasoning. To address this, we propose the context reward, which dynamically assigns rewards based on both the format of the answer and the length of the reasoning response. We define a satisfaction indicator $I_s$ to reflect the quality of the current answer. If the sum of $R_{syn}$, $R_{func}$ and $R_{div}$ is greater than a threshold $\Delta = 4$, then $I_s = 1$; otherwise, $I_s = 0$. Overall, the form of the reward function is as follows:

$$R_{cont} = \{ \begin{matrix} 0.5L_t + 0.5I_f & \text{if } I_s = 1 \\ -0.5L_t + 0.5I_f & \text{if } I_s = 0 \end{matrix} \tag{4}$$

where, $I_f$ indicates whether the output meets the required format. If so, it is assigned a value of 1; otherwise, it is assigned -1, $L_t$ represents the ratio of the average length of the chains of thought from the previous four responses to the length of the current chain of thought.

## 3 EXPERIMENTS

### 3.1 EXPERIMENTAL SETTINGS

**Dataset**.

Due to limited high-quality Verilog data, we built our dataset by combining open-source sources from GitHub and Hugging Face. After preliminary cleaning with Design Compiler, 5167 synthesis-passed code-description pairs were selected for Stage 1 (SFT). For Stage 2, 3570 natural language descriptions were extracted from another subset. Due to the scarcity of readily verifiable Verilog with testbenches, Stage 3 additionally incorporates the benchmark set Verilog-Eval (Liu et al., 2023b) into the training corpus. Its designs are of modest gate count and its natural-language descriptions are concise, which together tend to yield richer reward signals; in total, 829 functionally verified samples are included.

**Training Setting**.

We performed fine-tuning using GRPOTrainer, a reinforcement learning framework based on the `trl` library. To select a suitable base model, we ran pilot tests comparing different large language models on Verilog code generation. We ultimately chose Qwen 2.5-7B-Instruct-1M (abbreviated as Qwen 2.5) for its superior code-generation accuracy and output volume. All training was carried out on 8 NVIDIA A100 GPUs, each with 40 GB of memory. We adopt the LoRA method for fine-tuning (Hu et al., 2021), with the following hyperparameters: the rank of the LoRA layers is set to 4, the alpha parameter to 8, and the dropout rate to 0.1. During training, we use the Adam optimizer (Kingma & Ba, 2017) with a learning rate of 5e-5.

**Tesing Setting**.

We used all designs from the publicly available RTLLM v1.1 benchmark (Lu et al., 2023) as our test set. The temperature for LLM inference is set to 0.2.

**Evaluation Metrics**.

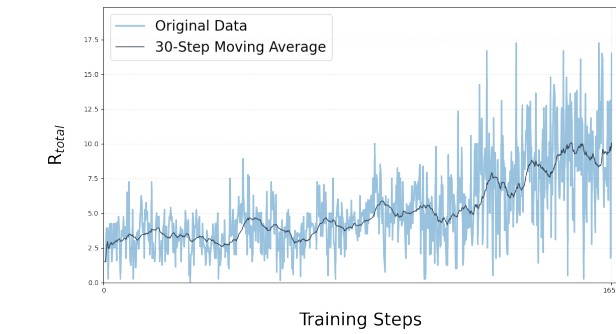

Figure 4: The trend graph of the total reward obtained by every training step.

We denote syntax correctness as Syn., functional correctness as Fun., generation performance as Gen., and evaluate the model performance based on the following aspects: (1) **OPMO_Pass@$k$**: The traditional pass@$k$ (Chen et al., 2021), referred to in this paper as **One-Prompt-One-Output pass@$k$** (OPOO_Pass@$k$), evaluates whether at least one of the Top-$k$ outputs is correct when each prompt generates a single Verilog module. To better assess the effectiveness of our approach, we introduce **One-Prompt-Multi-Output pass@$k$** (OPMO_Pass@$k$), which evaluates whether any correct result exists among the multiple Verilog modules generated for a single prompt. Specifically, we only consider the cases of $k = 1$ and $k = 5$. (2) **Success Rate:** We use the proportion of correct RTL module code across all generated modules obtained from five queries as the Success Rate, Table 2 show only functional Success Rate, detailed syntactical Success Rate could be found in Appendix A.2. (3) **Quantitative Metrics:** Gen.Num: The number of Verilog codes generated per prompt. Syn.Num: The number of syntactically valid codes per prompt. Fun.Num: The number of functionally correct codes per prompt.

## 3.2 EVALUATION RESULTS

**Training Dynamics of Reinforcement Learning**.

Figure 4 shows the original data and the 30-step moving average of reward in Stage 3, where the overall trend exhibits a clear upward movement. This suggests that as training progresses, the model's rewards consistently increase, demonstrating the effectiveness of the training method.

**Overall Evaluation on RTLLM**.

We evaluate RTLSeek on RTLLM v1.1 in Table 1 against both commercial foundation models–Qwen2.5-Instruct (Yang et al., 2025), GPT-4o (Achiam et al., 2023), and DeepSeek-R1 (Guo et al., 2025)–and open-source baselines, including the supervised-fine-tuned (SFT) DeepRTL (Liu et al., 2025), Thakur (Thakur et al., 2024), and RTLCoder (Liu et al., 2024), as well as RL post-training academic works like ChipSeek-R1 (Chen et al., 2025) and CodeV-R1 (Zhu et al., 2025). For foundation models we apply two prompt-engineering strategies: OPOO and OPMO. For the academic fine-tuned models (Thakur and RTLCoder) we report results with OPOO prompting and the corresponding OPOO_pass@k metric; OPMO prompting fails on these models because domain-specific fine-tuning degrades instruction-following ability—a manifestation of catastrophic forgetting that prevents generation of multiple correctly-formatted RTL modules. All other entries in the table are taken directly from the respective papers; however, there is every reason to expect that the same catastrophic-forgetting effect would occur in those models as well. Restricting the comparison to OPOO prompting is therefore fully justified. As shown in Table 1, RTLSeek raises the average functional success rate by 29% over Qwen2.5, despite using only 1% of R1's parameters. Its enhanced diversity yields more correct outputs per prompt, substantially improving OPMO_pass@5 and overall functional success rate.

## 3.3 ABLATION STUDY

**Study Settings and Model Descriptions**.

Table 1: Generate result comparison about different metrics.

| Type | Model | Params (#) | Syn.OPOO_pass@1 | Syn.OPOO_pass@5 | Fun.OPOO_pass@1 | Fun.OPOO_pass@5 |
|---|---|---|---|---|---|---|
| Foundation | Qwen 2.5-Instruct* | 7B | 0.48 | 0.71 | 0.27 | 0.48 |
| | GPT-4o* | / | 0.80 | 0.89 | 0.42 | 0.66 |
| | DeepSeek R1* | 671B | 0.77 | 0.86 | 0.55 | 0.73 |
| SFT models | DeepRTL-2 | 220M | 0.71 | 0.81 | 0.32 | 0.42 |
| | DeepRTL-1 | 16B | 0.74 | 0.77 | 0.38 | 0.35 |
| | RTLCoder* | 7B | 0.73 | 0.89 | 0.32 | 0.49 |
| | Thakur* | 16B | 0.83 | 0.86 | 0.17 | 0.24 |
| RL models | ChipSeek-R1 | 7B | / | 0.96 | / | 0.83 |
| | CodeV-R1 | 7B | / | / | 0.73 | 0.86 |
| Type | Model | Params (#) | Syn.OPMO_pass@1 | Syn.OPMO_pass@5 | Fun.OPMO_pass@1 | Fun.OPMO_pass@5 |
| Foundation | Qwen 2.5-Instruct* | 7B | 0.50 | 0.74 | 0.32 | 0.50 |
| | GPT-4o* | / | 0.86 | 0.93 | 0.5 | 0.71 |
| | DeepSeek R1* | 671B | 0.90 | 0.96 | 0.65 | 0.83 |
| Ours | RTLSeek* | 7B | **0.86** | **0.96** | **0.76** | **0.86** |

*Note:* Models marked with an asterisk (*) were evaluated by ourselves; all others are cited from the respective papers; bold denotes the best result except commercial LLMs; underline denotes the real best result.

To verify our hypotheses regarding which methods can effectively enhance the Verilog generation capabilities of LLMs, we conducted an ablation study. The ablated models we implemented include: (1) RTLSeek.w/o DR: RTLSeek without the Diversity Reward in Stage 2 and 3 RL training. (2) RTLSeek.w/o S3: RTLSeek without Stage 3 RL training, leaving only Stages 1 and 2. (3) RTLSeek.w/o S2: RTLSeek without Stage 2 RL training; after Stage 1, the process directly moved to Stage 3. (4) RTLSeek.only S1: RTLSeek only after Stage 1 OPOO SFT training.

**Ablation Study for Diversity Reward**.

As indicated in Table 2, RTLSeek outperforms RTLSeek. w/o DR in terms of Success Rate. This comparison suggests that diversity-oriented RL training is important for improving LLM-based RTL generation. In addition, RTLSeek also shows a significant increase in average Fun.OPMO_pass@5 when compared to RTLSeek. w/o DR. This improvement mainly stems from the diversity-oriented approach, which encourages the model to generate more varied responses per prompt, expanding the solution space and enhancing overall performance. We also conducted a more detailed study in Table 3. In Table 3, we divide the items of RTLLM v1.1 benchmark into two categories based on whether the model RTLSeek.w/o DR can pass: (1) w/o DR Pass: Items that RTLSeek.w/o DR can solve correctly at least once within five attempts. (2) w/o DR NO Pass: Items that RTLSeek.w/o DR can not solve (no correct solutions even after five attempts).

In Table 3, we report the average Success Rate of RTLSeek and RTLSeek.w/o DR on these two sets of items. (1) For items w/o DR NO pass, RTLSeek.w/o DR's average Success Rate is 0%, but Ours achieves 13.1%. (2) For items without DR pass, RTLSeek.w/o DR has 47% average Success Rate, while RTLSeek achieves 56%, surpassing Baseline by 19%. These results demonstrate that introducing a diversity reward not only boosts correctness on items that RTLSeek.w/o DR can already solve but also enables solving some RTLSeek.w/o DR unsolved items. Hence, diversity-oriented reinforcement learning training has a notable and measurable impact on enhancing LLM-based RTL code generation.

**Ablation Study for Multi-Stage RL Training**.

As indicated in Table 2, compared to RTLSeek.w/o S3, RTLSeek demonstrates a 166.7% improvement in average Success Rate. This indicates that Stage 3, by leveraging a training set equipped with testbenches for diversity-driven exploration and correctness-based learning, provides substantial benefits. Compared to RTLSeek.w/o S2, RTLSeek still demonstrates a 42.9% improvement in average Success Rate, with a significant boost in the number of codes generated. This improvement shows that the Stage 2 training facilitates better diversity capabilities and helps the model leverage the training set equipped with testbenches in Stage 3 for more effective exploration. The above evaluation and comparison demonstrate that, by incorporating a Diversity-Centric Multi-Objective Reward Scheduling and a three-stage training framework, RTLSeek outperforms GPT-4o in terms of RTL syntax and functional accuracy on academic benchmarks. Ablation studies further verify the significant effectiveness of the Diversity-Oriented Reinforcement Learning post-training paradigm.

**Summary**.

Table 2: Blation study on the diversity reward and multi-stage training.

| Ablation | Gen. Num | Syn. Num | Fun. Num | Syn.OPMO _pass@1 | Syn.OPMO _pass@5 | Fun.OPMO _pass@1 | Fun.OPMO _pass@5 | Success Rate |
|---|---|---|---|---|---|---|---|---|
| RTLSeek | 3.22 | **2.23** | **1.27** | **0.86** | **0.96** | **0.76** | **0.86** | **0.40** |
| RTLSeek.w/o DR | 1.70 | 0.85 | 0.54 | 0.64 | 0.73 | 0.55 | 0.59 | 0.35 |
| RTLSeek.w/o S3 | **4.14** | 1.31 | 0.61 | 0.41 | 0.73 | 0.32 | 0.55 | 0.15 |
| RTLSeek.w/o S2 | 2.42 | 1.33 | 0.67 | 0.55 | 0.77 | 0.45 | 0.59 | 0.28 |
| RTLSeek.only S1 | 1.15 | 0.32 | 0.17 | 0.32 | 0.59 | 0.23 | 0.31 | 0.15 |

Although RTLSeek does not produce the largest number of RTL implementations (falling short of the model without stage-3 training), it yields the highest proportion that pass both syntax and functional checks and achieves the highest overall module-level Success Rate, successfully striking a balance between generation quantity and quality.

Table 3: Fine-grained ablation study on two subsets obtained from the partitioning RTLLM v1.1 for Success Rate about the diversity reward.

| Model | w/o DR Pass | w/o DR NO Pass |
|---|---|---|
| RTLSeek | 0.56 | 0.131 |
| RTLSeek.w/o DR | 0.47 | 0 |

## 3.4 DISCUSSION WITH OTHER RL POST-TRAINING METHODS

Contemporary studies likewise adopt RL to boost LLM-based RTL generation. ChipSeek-R1 (Chen et al., 2025) erects a hierarchical reward pyramid in which EDA feedback–syntax, simulation, synthesis, Performance, Power and Area (PPA)–is stacked with strict dependencies: a model must first pass lower-level checks (e.g., compilable syntax) before it can harvest upper-level rewards (e.g., PPA improvement). CodeV-R1 (Zhu et al., 2025), in turn, focuses on data engineering, mass-producing high-quality samples via a "code $\rightarrow$ NL $\rightarrow$ code" round-trip synthesiser coupled with automatic testbench generation. Both methods, however, remain essentially imitation-based: exploration is incidental and the objective is to replicate already-known good implementations.

RTLSeek optimizes for diversity-driven exploration, helping the policy move beyond memorization to capture key hardware design abstractions, especially in data-scarce scenarios. By deferring strict correctness constraints, it better balances diversity and accuracy, offering a "one-spec, many-solutions" approach. This provides a diverse set of solutions in one pass, enabling faster, focused different oriented optimization where area or performance variants are prioritized over the absolute PPA optimum.

## 4 CONCLUSION

This paper presents RTLSeek, a novel post-training paradigm using Diversity-Oriented Reinforcement Learning to improve the accuracy and diversity of LLM-generated RTL. With a Diversity-Centric Multi-Objective Reward and a three-stage training framework, RTLSeek effectively addresses data scarcity. Experiments show it outperforms other methods on RTLLM benchmark, with ablation studies confirming its effectiveness.

## 5 REPRODUCIBILITY STATEMENT

To facilitate full reproduction of our results, we provide (i) the LoRA parameter weights of the RTLSeek model, the inference and evaluation code, as well as the dataset used for training, are all available at `https://anonymous.4open.science/r/ICLR24081-9C28/`. (ii) detailed prompt engineering design, RTL design human language description and case study are in Appendix A.3.

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

# A APPENDIX

## A.1 AST-BASED DIVERSITY ANALYSIS

To determine whether Verilog code segments differ substantively or merely in superficial aspects like variable naming or statement reordering, we implement equivalence verification based on AST, hierarchical representations that capture the syntactic structure of code while abstracting away lexical details such as whitespace and comments. For Verilog hardware description language, these tree structures effectively preserve module definitions, port declarations, signal assignments, and control structures in a form conducive to programmatic analysis (Takamaeda-Yamazaki, 2015b).

We parse source code into ASTs using pyverilog, then apply a recursive comparison algorithm that performs a depth-first traversal of both trees as shown in Figure 5. This algorithm methodically compares node types, attributes, and structural relationships to determine complete equivalence. By operating at the structural level rather than the textual level, our approach overcomes limitations of text-based comparison by accurately identifying structural equivalence despite surface differences.

When code segments differ only in variable names (e.g., "p" versus "q"), our system recognizes their equivalence, while correctly distinguishing functionally different implementations even when they share certain syntactic patterns. This recursive verification system supports diversity exploration in RTL code generation for RTL-Seek by filtering out superficially different but structurally identical variants.

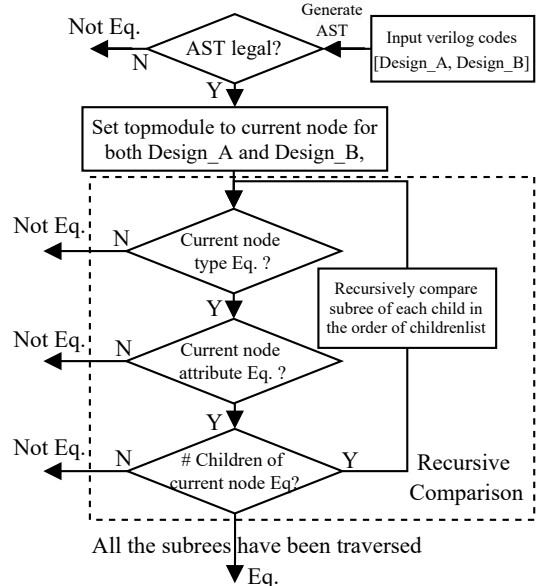

Figure 5: Verilog structural equivalence testing algorithm based on abstract syntax tree.

## A.2 ADDITIONAL EXPERIMENT RESULTS

**Evaluation on Selected Items from RTLLM.**

To facilitate a more fine-grained comparison, we selected specific tasks and provided a detailed presentation and comparison in Table 4. We selected only the designs for which the unfinetuned Qwen 2.5 model achieved a low OPMO_pass@5 syntactic accuracy, in order to highlight RTLSeek's improvement on challenging cases.

In Table 4, we report the results across five outputs of the model for each task, and then summarize the average Success Rate and OPMO_pass@5 metrics for different models on these tasks. Each cell in Table 4 lists the correctness ratio over five attempts. The cells highlighted in green indicate better performance, yellow indicate intermediate performance, and red indicate lower performance. The comparison results in Table 4 show that RTLSeek outperforms the advanced models R1 and GPT-4 in terms of average Success Rate and OPMO_pass@5.

Interestingly, the comparison in Table 4 shows that DeepSeek R1 demonstrates a higher accuracy in the items that it can solve correctly. For the given question, if it can generate a correct answer, its OPMO_pass@5 is nearly always 100% or 80% (e.g., multi_booth_8bit, comparator_4bit, adder_bcd). On the other hand, RTLSeek does not always guarantee correct generations for the designs it has solved before. However, compared with R1, our model is able to correctly generate solutions for a wider range of designs, such as LFSR, freq_divbyfrac, sequence_detector. On these tasks, R1 struggles to arrive at the correct answers. A possible explanation for this difference is that, as a larger model, R1 exhibits superior stability. Meanwhile, RTLSeek benefits from diversity-oriented reinforcement learning training, resulting in enhanced diversity.

**Ablation Study about Selected Items from RTLLM.**

For the ablation study models, we conduct a more detailed experimental analysis using the same set of items selected from the RTLLM benchmark as shown in Table 4. This analysis aims to demonstrate the effectiveness of RTLSeek's Diversity-Centric Multi-Objective Reward Scheduling and the three-stage post-training fine-tuning strategy.

- RTLSeek.w/o DR: RTLSeek without the Diversity Reward in Stage 2 and 3 RL training.
- RTLSeek.w/o S3: RTLSeek without Stage 3 RL training, leaving only Stages 1 and 2.

Table 4: The accuracy of successful attempts out of five or all trials for RTLSeek in generating Verilog compared with other models, assessing syntax correctness (Syn.) rate (%) and functional correctness (Fun.) rate (%).

| Design | GPT-4 | | Deepseek R1 | | Qwen 2.5 | | RTLSeek | |
|---|---|---|---|---|---|---|---|---|
| | Syn.(%) | Fun.(%) | Syn.(%) | Fun.(%) | Syn.(%) | Fun.(%) | Syn.(%) | Fun.(%) |
| fixed_point_adder | 80 | 80 | 80 | 80 | 20 | 0 | 80 | 80 |
| comparator_3bit | 0 | 0 | 100 | 80 | 60 | 60 | 80 | 80 |
| multi_booth_8bit | 60 | 60 | 100 | 80 | 60 | 60 | 80 | 80 |
| comparator_4bit | 40 | 40 | 100 | 100 | 80 | 20 | 100 | 60 |
| adder_bcd | 0 | 0 | 100 | 80 | 0 | 0 | 40 | 40 |
| LFSR | 0 | 0 | 0 | 0 | 0 | 0 | 60 | 20 |
| freq_divbyodd | 60 | 0 | 80 | 0 | 60 | 0 | 80 | 40 |
| freq_divbyfrac | 40 | 0 | 100 | 0 | 0 | 0 | 80 | 20 |
| freq_divbyevn | 0 | 0 | 0 | 0 | 0 | 0 | 40 | 0 |
| fixed_point_substractor | 20 | 20 | 60 | 60 | 0 | 0 | 80 | 60 |
| sequence_detector | 0 | 0 | 0 | 0 | 0 | 0 | 40 | 40 |
| float_multi | 100 | 60 | 80 | 40 | 0 | 0 | 100 | 40 |
| **Success Rate** | **33.3** | **21.7** | **66.7** | **43.4** | **25** | **5** | **71.7** | **46.7** |
| **OPMO_pass@5** | **58.3** | **41.7** | **75** | **58.3** | **41.7** | **25** | **100** | **91.7** |

- RTLSeek.w/o S2: RTLSeek without Stage 2 RL training; after Stage 1, the process directly moved to Stage 3.

- RTLSeek.only S1: RTLSeek only after Stage 1 OPOO SFT training.

Table 5 presents the syntax correctness (Syn.) and functional correctness (Fun.) rates (%) of RTLSeek compared to ablation study models, measuring the accuracy of successful attempts out of five or all generated trials. The results show that, in terms of overall syntax and functional success rates across all generated RTL modules, the ablation models perform similarly but consistently fall short of the full RTLSeek model, which integrates both Diverse Reward and the three-stage post-training framework. Notably, under the OPMO_pass@5 metric, the model trained solely with the first-stage SFT performs significantly worse than those further fine-tuned via GRPO, highlighting the effectiveness of reinforcement learning for RTL generation, though still not reaching the performance of RTLSeek. RTLSeek without the Diversity Reward in Stage 2 and 3 reinforcement learning achieves the best performance metrics against other GRPO training models, which suggests that the model underwent extensive fine-tuning compared with other models.

**Temperature Analysis of RTLSeek Temperature Evaluation on RTLLM**.

To evaluate the influence of sampling temperature on the quality of RTL code generation, we conduct a systematic analysis of RTLLM across different temperature values ranging from 0.2 to 0.8. As shown in Figure 6, we report both syntactic and functional correctness under the **One-Prompt-Multi-Output** (OPMO) evaluation protocol. The results demonstrate several key patterns. First, the syntactic correctness remains consistently high across all temperature settings, with **Syntax OPMO_pass@5** maintaining over 90% accuracy. However, there is a noticeable downward trend in **Syntax OPMO_pass@5** as temperature increases, suggesting that higher sampling temperatures lead to more variation at the cost of reduced deterministic correctness. More importantly, the **functional correctness** metrics are significantly more sensitive to temperature variation. While **functional OPMO_pass@5** consistently outperforms **pass@1**, both metrics decline notably beyond a temperature of 0.2. This is particularly evident in **functional OPMO_pass@1**, which drops from approximately 75% to below 70% as temperature increases. These findings indicate that, for RTL generation—a task where structural precision and determinism are critical—higher temperatures introduce harmful randomness that outweighs the potential benefits of output diversity.

Table 5: The accuracy of successful attempts out of five or all trials for RTLSeek (Table 4) in generating Verilog compared with ablation study models, assessing syntax correctness (Syn.) rate (%) and functional correctness (Fun.) rate (%).

| Design | RTLSeek.w/o S2 | | RTLSeek.w/o S3 | | RTLSeek.w/o DR | | RTLSeek.only S1 | |
|---|---|---|---|---|---|---|---|---|
| | Syn.(%) | Fun.(%) | Syn.(%) | Fun.(%) | Syn.(%) | Fun.(%) | Syn.(%) | Fun.(%) |
| fixed_point_adder | 40 | 20 | 20 | 20 | 100 | 100 | 0 | 0 |
| comparator_3bit | 80 | 40 | 80 | 80 | 80 | 80 | 60 | 20 |
| multi_booth_8bit | 20 | 20 | 20 | 20 | 80 | 80 | 0 | 0 |
| comparator_4bit | 20 | 20 | 20 | 20 | 60 | 20 | 40 | 0 |
| adder_bcd | 0 | 0 | 0 | 0 | 60 | 20 | 0 | 0 |
| LFSR | 0 | 0 | 0 | 0 | 0 | 0 | 0 | 0 |
| freq_divbyodd | 40 | 0 | 100 | 0 | 40 | 0 | 60 | 0 |
| freq_divbyfrac | 100 | 0 | 80 | 0 | 80 | 0 | 60 | 0 |
| freq_divbyevn | 0 | 0 | 0 | 0 | 0 | 0 | 0 | 0 |
| fixed_point_substractor | 20 | 20 | 20 | 20 | 40 | 20 | 20 | 20 |
| sequence_detector | 0 | 0 | 0 | 0 | 0 | 0 | 0 | 0 |
| float_multi | 20 | 0 | 0 | 0 | 80 | 0 | 20 | 0 |
| **Success Rate** | **33.3** | **25** | **36.4** | **14.7** | **48.9** | **31.3** | **21.6** | **14.3** |
| **OPMO_pass@5** | **66.7** | **41.7** | **58.3** | **41.7** | **75** | **50** | **50** | **16.7** |

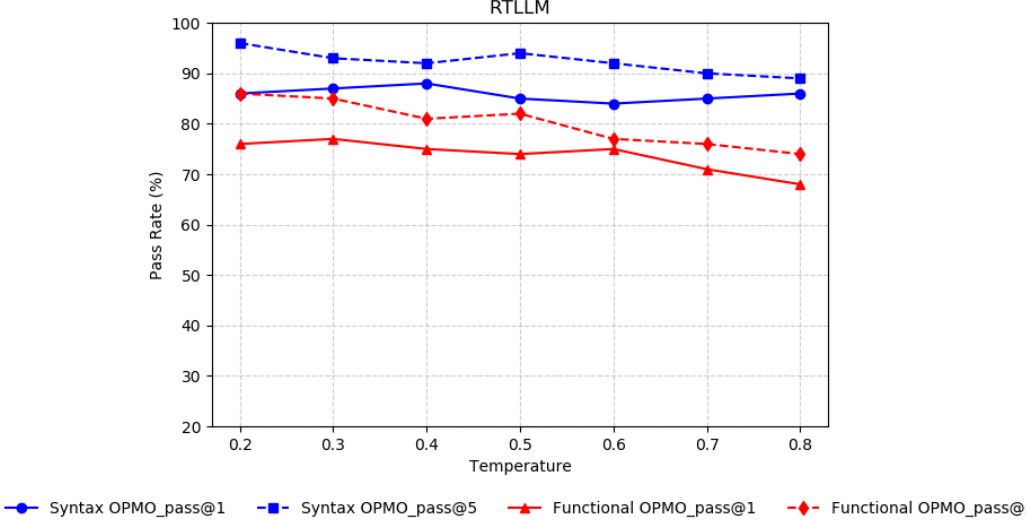

Figure 6: Temperature analysis of RTLSeek.

## A.3 IMPLEMENTATION DETAILS

**Prompt Engineering**.

The prompt engineering required for evaluation on the RTLLM benchmark is outlined in the following list, including both *system prompts* and *user prompts*. The *user prompt* can be further divided into several components: *creative thinking prompt*, *syntax prompt*, and *generate more prompt*.

[System Prompt]

You are an accomplished expert in integrated circuit design, tasked with generating RTL code based on natural language specifications.

Respond in the following format:

$< think >$

...

$< /think >$

$< total\_design >$

...

$< /total\_design >$

...

$< total\_design >$

...

$< /total\_design >$

[User Prompt]

Creative Thinking Prompt + RTL Design Human Language Description + Syntax Prompt + Generate More Prompt.

[Creative Thinking Prompt]

To ensure comprehensive reasoning and accurate responses, please carefully analyze the following query.

Break down the problem into its fundamental components, evaluate potential solutions, and provide a detailed, step-by-step explanation of your thought process.

Avoid skipping any critical thinking steps or providing premature conclusions.

Your response should reflect a deep understanding of the topic and demonstrate logical coherence throughout.

Place your thought process between ¡think¿ and ¡/think¿

[Syntax Prompt]

You should note that in Verilog, signals of type wire cannot be assigned directly in the always block.

A wire type signal is usually used for continuous assignment (assign statement) or the output of a module.

A signal of type reg can be assigned a value in the always block.

It is important to note in the design that undefined modules need to be defined first.

[Generate More Prompt]

Generating as much structurally diverse Verilog code as possible can be optimized from the perspective of area, power, and minimizing the worst path delays.

The generated Verilog code should be placed between $< total\_design >$ and $< /total\_design >$, and the generated code optimized from different angles should be placed in different $< total\_design >$ and $< /total\_design >$ tags.

Respond in the following format:

$< think >$

...

$< /think >$

$< total\_design >$

...

$< /total\_design >$

...

$< total\_design >$

...

$< /total\_design >$

...

The code for each design should have only one module, and you need to avoid multiple modules in a design.

You should strictly abide by my requirements and restrictions.

**RTL Design Human Language Description**.

In our dataset, both the training and evaluation sets share a consistent data format, which is closely aligned with the nature of our task—translating human natural language circuit specifications into RTL code. To facilitate this translation, the natural language descriptions must be appropriately constrained and standardized. Taking an adder design from the RTLLM benchmark as an example, the description typically includes the following components: (1) a brief summary of the design objective, such as specifying the development of a 16-bit adder; (2) the designated module name; (3) a detailed functional and structural description of each input and output port defined in the module; and (4) a description of the internal logic implementation of the module.

[RTL Design Human Language Description Example of a 16-bit full adder]

Implement a module of a 16-bit full adder in combinational logic.

Module name: adder_16bit

Input ports:

a[15:0]: 16-bit input operand A.

b[15:0]: 16-bit input operand B.

Cin: Carry-in input.

Output ports:

y[15:0]: 16-bit output representing the sum of A and B.

Co: Carry-out output.

Implementation:

In the adder_16bit module, you need to design a small bit-width adder (8-bit adder), which will be instantiated multiple times.

Give me the complete code.

## A.4  CASE STUDY

To illustrate the distinct generation capabilities of RTLSeek, we conduct a case study on the following circuit design *ring_counter*. The natural language specification of the circuit and the corresponding RTL Verilog code generated by the model are presented below.

*ring_counter_1* utilizes a shift-based implementation where the ring counter state is updated by left-shifting the register and inserting the inverted value of the most significant bit (MSB) into the least significant bit (LSB) position. While this approach is compact, it introduces logical errors that disrupt the intended cyclic behavior of a true ring counter. In contrast, *ring_counter_2* adopts a finite state machine (FSM) approach, explicitly enumerating all valid states using a case statement and defining their deterministic transitions. This ensures precise control over state evolution and guarantees correct circular progression through all eight states. Therefore, *ring_counter_2* offers superior correctness and reliability at the expense of greater structural complexity compared to the more concise but flawed logic in *ring_counter_1*.

---

Implement a module of a parameterized fixed-point adder for arithmetic operations with fixed precision.

Implement a module of an 8-bit ring counter for cyclic state sequences.

Module name:

ring_counter

Input ports:

clk: Clock signal that drives the state transitions of the ring counter.

reset: Reset signal to initialize the counter to its starting state.

Output ports:

out [7:0]: 8-bit output representing the current state of the ring counter. Only one bit is set high at any time, and the set bit cycles through the 8-bit output.

Internal logic:

1. State Transition: The ring counter follows a cyclic pattern where exactly one bit is set to 1 in the output at any given time, and the 1 shifts to the next bit with each clock pulse.

2. Initialization: When the reset signal is high, the counter is initialized to its starting state, typically with the least significant bit (LSB) of out set to 1 (i.e., out = 8'b0000_0001).

3. Cycling Behavior: On each rising edge of the clock signal, the 1 shifts to the next bit in the sequence, and after reaching the most significant bit (MSB), it wraps around to the LSB, creating a cyclic sequence.

Reset Behavior: When reset is high, the ring counter is reset to its initial state (out = 8'b0000_0001).

Give me the complete code.

---

*< think >*

To implement an 8-bit ring counter in Verilog, we need to follow these steps:

1. Define the necessary logic for the state transition.

2. Use a register to store the current state.

3. Shift the bits of the state on each rising edge of the clock.

4. Ensure that the reset signal initializes the counter to the correct starting state.

5. Make sure the design is structurally diverse and optimized.

We will use a single module with a reg array to store the state and an always @(posedge clk) block to handle the state transition. We will also define the reset behavior explicitly.

*< /think >*

---

```
< total_design >
module ring_counter_1 (
input clk,
input reset,
output reg [7:0] out
);
// Initialize the counter to the starting state when reset is high
always @(posedge clk or posedge reset) begin
if (reset)
out <= 8'b0000_0001;
else
out <= out[6:0],  out[7];
end
endmodule
< /total_design >
```

```
< total_design >
module ring_counter_2 (
input clk,
input reset,
output reg [7:0] out
);
reg [7:0] state;
// Initialize the counter to the starting state when reset is high
always @(posedge clk or posedge reset) begin
if (reset)
state <= 8'b0000_0001;
else
state <= state[6:0], state[7];
end
// Alternative method using a case statement for state transition
always @(posedge clk) begin
case (state)
8'b0000_0001: state <= 8'b0000_0010;
8'b0000_0010: state <= 8'b0000_0100;
8'b0000_0100: state <= 8'b0000_1000;
8'b0000_1000: state <= 8'b0001_0000;
8'b0001_0000: state <= 8'b0010_0000;
8'b0010_0000: state <= 8'b0100_0000;
8'b0100_0000: state <= 8'b1000_0000;
8'b1000_0000: state <= 8'b0000_0001;
default: state <= 8'b0000_0001;
endcase
end
assign out = state;
endmodule
< /total_design >
```

