# OpenReview forum: "RTLSeek: Boosting the LLM-Based RTL Generation with Diversity-Oriented Reinforcement Learning"
_ICLR.cc/2026/Conference — Submitted to ICLR 2026_

### Official Review · Reviewer_K9ne · 2025-10-27

**Soundness:** 2
**Presentation:** 3
**Contribution:** 2
**Rating:** 2
**Confidence:** 4

**Summary:**

The paper introduces RTLSeek, a framework that improves how large language models generate hardware designs (RTL/Verilog) from natural language by making them more accurate and diverse. It uses multi-objective rewards (syntax, functionality, diversity) and a three-stage finetuning process to make the model explore multiple valid implementations despite limited data.

**Strengths:**

- Introduces a novel diversity-oriented reinforcement learning framework for LLM-based RTL generation, combining correctness and structural diversity.
- Methodology is technically sound, with a well-defined three-stage training process and comprehensive ablation studies supporting design choices.

**Weaknesses:**

- Despite the methodological novelty, RTLSeek achieves performance comparable to prior RL-based methods (e.g., ChipSeek-R1, CodeV-R1). The results do not clearly demonstrate a substantial improvement that justifies the added complexity of the framework.
- No analysis of why improvements are small.
- There is no analysis of how the diverse RTL implementations perform under standard EDA synthesis flows, nor a clear demonstration of the practical benefits of high diversity.

**Questions:**

- The improvements over prior RL-based methods (ChipSeek-R1, CodeV-R1) are relatively small. Can the authors provide insights or additional analysis explaining why the gains are limited despite the richer reward design?
- The paper mentions using an AST-based structural analysis to quantify diversity but does not specify the threshold or metric used to determine when two designs are considered distinct. Could the authors provide examples of acceptable versus unacceptable diverse RTL generations, along with clear criteria illustrating how diversity scores are computed?
- The paper mentions a large dataset used across the three training stages, but it does not describe its diversity in terms of design complexity (e.g., gate count, wire count, or functional difficulty). Could the authors provide statistics that characterize the dataset’s variety?
- The paper states that functional correctness is one of the reward objectives, but it is unclear how correctness is verified during evaluation. Some of the provided testbenches appear not to be self-checking. Could the authors clarify how they ensure that the generated RTL designs are functionally correct (e.g., through automated comparison or manual inspection)?

---

### Official Review · Reviewer_QG1N · 2025-10-29

**Soundness:** 2
**Presentation:** 1
**Contribution:** 2
**Rating:** 4
**Confidence:** 4

**Summary:**

This paper presents RTLSeek, a novel method that significantly improves the diversity and quality of LLM-generated RTL code. By employing a multi-stage, diversity-oriented RL framework with AST-based diversity quantification and GRPO optimization, it addresses key limitations in existing post-training methods. On the RTLLM v1.1 benchmark, it outperforms strong baselines, enhancing Qwen 2.5's RTL generation by over 40% and functional success rate by 29%. The implementation is publicly available.

**Strengths:**

1. This paper addresses the core challenges in the field of RTL generation: the scarcity of high-quality, verifiable data and the lack of design diversity in existing LLM post-training, by presenting a logically sound solution.
2. The paper's methodology, which reasonably combines SFT with a two-stage GRPO process for applying LLMs to RTL generation, is sound and well-motivated.
3. The experimental setup is well-designed. The results demonstrate significant improvements over the baseline, and the provision of an anonymized link ensures reproducibility.

**Weaknesses:**

1. The GRPO algorithm itself is not a fundamentally new contribution (it is an improvement derived from DeepSeek-R1), and multi-stage training has precedents in LLM post-training. Therefore, the innovation of this work lies more in its technical adaptation to the RTL domain than in a breakthrough at the underlying algorithmic level.
2. The study exhibits limited generalizability of the base model. It is validated only on Qwen 2.5, without testing other mainstream LLMs mentioned in the paper, such as Llama, GPT-4o, or Deepseek. This failure to demonstrate the method's adaptability across different base models, combined with the lack of experimentation on LLMs of varying scales, significantly weakens the claimed robustness.
3. The paper fails to specify the scale of the problems addressed in RTL generation. Please provide detailed information regarding the data, such as its scale, at least in the supplementary materials.
4. The work fails to investigate the impact of problem scale on the method's effectiveness, such as its viability for extremely large-scale problems. Additionally, it does not differentiate the performance between combinational logic (e.g., adders) and sequential logic (e.g., counters), and omits analysis of how the degree of data scarcity (e.g., performance in extreme few-shot scenarios) affects the results.
5. Minor suggestions on formatting: The current manuscript's readability could be enhanced. If possible, please consider adjusting the layout. For instance, using \item for multiple numbered items within a paragraph (such as the contributions listed in the introduction), and adjusting Figure 1 (increasing the font size) and Figure 4 (the margins on both sides). These changes would improve the paper's readability.

**Questions:**

1. Please respond to the concerns I have raised in the 'weaknesses' section. If the revision can adequately address most of the critical issues, I would consider raising the score.

---

### Official Review · Reviewer_Dpxb · 2025-11-04

**Soundness:** 3
**Presentation:** 3
**Contribution:** 2
**Rating:** 2
**Confidence:** 4

**Summary:**

This paper introduces RTLSeek, a post-training framework designed to improve the Register Transfer Level (RTL) Verilog code generation capabilities of Large Language Models (LLMs). The core methodology involves a multi-stage, diversity-oriented reinforcement learning approach. It combines Supervised Fine-Tuning (SFT) with a reinforcement learning paradigm based on Group Relative Policy Optimization (GRPO). The authors design a multi-objective reward function to balance the correctness and diversity of the generated code. Experimental results on the RTLLM benchmark reportedly show that RTLSeek outperforms several baseline models.

**Strengths:**

+ Timely topic
+ Important attempt on RTL generation.

**Weaknesses:**

While the paper addresses the important and challenging problem of automating RTL design with LLMs, it suffers from significant flaws in its methodology, justification, and practical contribution. In its current form, it is not ready for publication.
1. The Contribution is Minor and Lacks Methodological Novelty.
The primary technical contribution of this paper is the application of the existing Group Relative Policy Optimization (GRPO) algorithm to the domain of Verilog code generation. While applying an existing algorithm to a new domain can be a valid contribution, this paper fails to sufficiently justify the novelty or necessity of this application. GRPO itself is not a novel algorithm proposed by the authors, and the paper does not articulate any unique challenges in RTL generation that required significant or non-trivial modifications to the GRPO framework. The work primarily consists of plugging an off-the-shelf RL algorithm into a new problem space, which makes the paper read more like a technical report on an application rather than a methodological breakthrough.
2. The Core "Multi-Objective Reward" Method is Critically Under-discussed.
The multi-objective reward mechanism (R_total = R_syn + R_func + R_div + R_cont) is the cornerstone of the paper's claims about balancing diversity and correctness, yet its treatment is extremely superficial. The authors simply list the components of the reward function without a deep discussion of key questions:
• Weighting and Trade-offs: How were the relative weights of these reward components determined? Is there a conflict between them (e.g., does optimizing for diversity harm functional correctness)? The paper provides no ablation studies or sensitivity analysis on the reward weights.
• Design Rationale: Is the reward function designed based on heuristics, or is there a deeper theoretical motivation? For example, the formulation of the diversity reward R_div as Nc + Ns seems overly simplistic and may not effectively guide the model to produce meaningful structural differences over superficial syntactic ones.
• Dynamic Scheduling: The title and abstract highlight "Scheduling," but this concept is vaguely explained in the main body. How is the reward mechanism "scheduled" across the three training stages? How do the priorities shift between stages? The absence of these critical details makes this central part of the methodology unconvincing.
3. The Paper Has Severe Reproducibility Issues.
The authors claim to provide code and model weights in the supplementary material, but what has been submitted is grossly incomplete.
• Incomplete Code: The provided "source code" consists of only a single Python file. This is entirely insufficient for reproducing the complex three-stage training pipeline, which involves simulation, verification, and evaluation. A complete repository should include scripts for data preprocessing, model definitions, training, evaluation, and environment dependencies.
• Missing Model Weights: The model weights, which are essential for verifying the performance claims of the paper, have not been provided.
The lack of complete, runnable code and model weights makes it impossible for the community to independently verify the paper's results, which is a major flaw in modern machine learning research.
4. The Fundamental Premise of Focusing on Diversity is Questionable.
The paper elevates "diversity" to a level of importance nearly equal to "correctness," but this core premise is not well-justified for this specific task.
• Confusion of Task Objective: Unlike creative tasks like writing, the ultimate goal of code generation is to provide the user with a single piece of code that is correct, efficient, and usable. While diversity can be a useful mechanism during training to encourage exploration and escape local optima, this paper treats diversity as an end goal in itself.
• Lack of Justification from a Practical Standpoint: From a user's perspective, for a given design specification, they typically need the single best implementation (e.g., optimal in terms of Performance, Power, and Area - PPA). The paper fails to argue convincingly why a user would need a model to return multiple, structurally different but functionally identical Verilog implementations at inference time. It is not clear if this diversity translates to tangible benefits, such as offering different PPA trade-offs. The evaluation metrics do not capture this, making the pursuit of diversity seem disconnected from the practical needs of RTL design.

**Questions:**

see above

---

### Official Review · Reviewer_dPL3 · 2025-11-10

**Soundness:** 2
**Presentation:** 3
**Contribution:** 2
**Rating:** 4
**Confidence:** 5

**Summary:**

This paper proposes RTLSeek, a post-training paradigm to improve the accuracy and diversity of LLM for RTL code generation. The authors employ multi-stage training with multi-objective reward scheme incorporating syntax, functionality and diversity metrics derived from AST analysis. Experiments on RTLLM benchmark demonstrate improved syntactic and cuntional accuracy.

**Strengths:**

(1) The paper propose a clear three-stage pipeline of integrating SFT and RL.

(2) Incorporate AST-based structural diversity as a metric.

**Weaknesses:**

(1) The conceptual noverlty is incremental, mainly relying on prior RL for code paradigms. GRPO adaptation offers minimal conceptual contribution beyond existing techniques

(2) Evaluation lacks rigor: unclear statistical significance. It seems RL was conducted on VerlogEval benchmarks and evaluation done on RTLLM. Given the scarcity of data, the paper did not provide solution to the data challenge and instead uses a well-know valuable benchmark for training instead.

(3) Details for OPOO and OPMO is not well clarified. Specifically I don't really understand why OPMO is used at all (as it does not seem to be a standard eval setting for code generation?).

**Questions:**

(1) For Table 2 comparisions, are the comparisons between different methods using the same number of training steps in RL? A more detailed settings config would make sure if the comparisons are fairly conducted.

(2) Why use VerilogEval as part of the training data for Stage 3 RL training? It is weird that a benchmark is used for training.

---

### Meta-Review · Area_Chair_4RTt · 2026-01-07

**Summary:**

The paper studies using Large Language Models (LLMs) for generating Register Transfer Level (RTL) code. The main idea in the paper is to post-train using standard policy gradient (GRPO) algorithm to incentivize the model to generate structurally diverse solutions. The key argument is that a multi-objective reward system, balancing syntax, functionality, and Abstract Syntax Tree (AST) diversity, can improve the model's ability to explore the design space and generate correct hardware logic.

Reviewers unanimously noted that the method is mostly an application of the existing GRPO algorithm to this application domain. I imagine this work is more suited for a electronic design automation venue. It was also pointed out that even with the complex training pipeline, the improvements over prior RL-based methods (like ChipSeek-R1) are relatively small. Therefore, I recommend rejecting the paper.

**Reviewer Concerns:**

No rebuttal was submitted.

**Reviewer Scores:**

No rebuttal was submitted.

---

### Decision · Program_Chairs · 2026-01-26

Reject